# Codling Moth Wing Morphology Changes Due to Insecticide Resistance

**DOI:** 10.3390/insects10100310

**Published:** 2019-09-21

**Authors:** Ivana Pajač Živković, Hugo Alejandro Benitez, Božena Barić, Zrinka Drmić, Martina Kadoić Balaško, Darija Lemic, Jose H. Dominguez Davila, Katarina Maryann Mikac, Renata Bažok

**Affiliations:** 1Department for Agricultural Zoology, Faculty of Agriculture, University of Zagreb, 10000 Zagreb, Croatiabaric@agr.hr (B.B.); zrinka.drmic@gmail.com (Z.D.); mmrganic@agr.hr (M.K.B.) rbazok@agr.hr (R.B.); 2Departamento de Biología, Facultad de Ciencias, Universidad de Tarapaca, Arica 1000000, Chile; hbenitez@uta.cl; 3South Coast Structural Engineers, P.O. Box U9 Wollongong, NSW 2500, Australia; jose_d@scsengineers.com.au; 4Centre for Sustainable Ecosystem Solutions, Faculty of Science, Medicine and Health, School of Earth, Atmospheric and Life Sciences, University of Wollongong, Wollongong, NSW 2522, Australia; kmikac@uow.edu.au

**Keywords:** geometric morphometrics, finite element method, forewing shape, biomarker

## Abstract

The codling moth (CM) (*Cydia pomonella* L.) is the most important apple pest in Croatia and Europe. Owing to its economic importance, it is a highly controlled species and the intense selection pressure the species is under has likely caused it to change its phenotype in response. Intensive application of chemical-based insecticide treatments for the control of CM has led to resistance development. In this study, the forewing morphologies of 294 CM (11 populations) were investigated using geometric morphometric procedures based on the venation patterns of 18 landmarks. Finite element method (FEM) was also used to further investigate the dispersal capabilities of moths by modelling wing deformation versus wind speed. Three treatments were investigated and comprised populations from integrated and ecological (susceptible) orchards and laboratory-reared non-resistant populations. Forewing shape differences were found among the three treatment populations investigated. Across all three population treatments, the movement of landmarks 1, 7, 8, 9, and 12 drove the wing shape differences found. A reliable pattern of differences in forewing shape as related to control practice type was observed. FEM revealed that as wind speed (m/s^−1^) increased, so too did wing deformation (mm) for CM from each of the three treatments modelled. CM from the ecological orchards displayed the least deformation followed by integrated then laboratory-reared CM, which had the highest wing deformation at the highest wind speeds. This study presents an affordable and accessible technique that reliably demonstrates wing shape differences, and thus its use as a population biomarker to detect resistance should be further investigated.

## 1. Introduction

The codling moth (CM) (*Cydia pomonella* L.) is the most important apple pest from the family Tortricidae causing economic losses in fruit production in Europe and globally [1,2,3,4]. Damage to fruit, caused by larval feeding, results in quality and yield losses, therefore, the establishment of effective plant protection measures is a major management issue in apple growing. The pest is endemic to Eurasia, however, during the last two centuries it has dispersed globally with the cultivation of apples and pears [3]. Over 70% of insecticide treatments used in commercial apple orchards are currently applied to control CM populations. CM is highly adaptable to different climatic and management conditions, and it is assumed that the pest has developed many ecotypes based on differing agro-ecological requirements [5,6]. Intensive application of insecticide treatments for the control of CM in the past lead to resistance development to various chemical groups of insecticides, and the diversity of insecticide resistance mechanisms has been well explored in European and worldwide CM populations [7,8,9].

The first case of insecticide resistance in CM was recorded in 1928 on arsenates in the United States [7,10]. In Europe, CM resistance to insecticides occurred in the 1990s, with the emergence of the first resistant populations to diflubenzuron in Italy and southeastern France [11,12], Switzerland, and Spain [7]. Currently, the resistance spectrum of CM populations in the whole of Europe has increased dramatically, and now include resistance to benzoylureas; benzoylhydrazines; neonicotinoids; organophosphates; macrocyclic lactones; pyrethroids; and eco-friendly avermectins [7,8].

An additional problem has emerged via the development of cross resistance, where CM have simultaneously developed resistance to several chemical subgroups of insecticides [13,14,15]. In more recent studies, special attention was given to determining the mechanisms of resistance of CM to certain chemical groups of insecticides. To assist with this endeavor, studies of the genetic structure and control strategies of CM populations were undertaken. Franck et al. [3] were the first to use microsatellites [16,17] to estimate genetic structure in CM populations from France. They found low genetic differentiation among populations and a marginal impact of insecticide treatments on the allelic richness of CM. These findings were corroborated by others from France [18] and Chile [19] who found low genetic variation in CM populations. In Croatia, low genetic differentiation among CM populations was also found, however, there was significant partitioning of genetic variation within individuals (70–98%) between treated and non-treated CM populations [20]. The lack of genetic differentiation found between populations along with the significant partitioning of genetic variation within individuals from CM populations under differing control systems indicates that genetic changes are occurring. However, the marker is unable to discern historical from contemporary changes. It is the contemporary variation in the genotype or phenotype that is most useful to understand and quantify in order to undertake successful integrated pest management (IPM). The discrepancies between historical and contemporary changes in genotype and phenotype have been extensively studied in other insect species, with workers suggesting phenotypic markers be used over genotypic markers when investigating rapidly evolving insect pest species that are often subject to intense selection pressure from control practices (e.g., western corn rootworm [21,22,23]).

Indeed, during the last decade, a significant change in the biology of CM was observed in the field. That is, CM now emerges earlier in the spring, the flight of moths lasts longer in the autumn, and the total number of moths caught during the vegetation season has increased [24]. Recent studies of CM biology and ecology in Croatia have revealed that CM develops two generations per year in ecological apple production systems (i.e., without use of chemical agents), with a third CM generation found in integrated apple production systems [25]. Such important data on their changed life history potentially indicates that different management types of production can affect the biology of CM. Yet, population genetic studies performed on the same CM populations, which were grouped according to management type, revealed no significant differences in genetic structure [20], suggesting that alternative biomarkers are needed to detect population-based differences [21,26].

Recent advances in geometric morphometric techniques (i.e., shape analysis) have enabled the quantification of insect morphology (size and shape) to investigate population differences [21,22,23,27,28,29,30,31]. It has been shown that metric properties such as wing shape and size are the first morphological characters to change, as these adult traits reflect the influence of both environmental and genetic factors [26,32]; thus, making them an ideal tool to detect and monitor population variation and resistant variants [21,23]. By analyzing the body shape of the target species, it is possible to determine whether physical traits have changed according to prevailing habitat characteristics. Further to the use of geometric morphometrics as a monitoring tool, it is possible to also gain important information on the basic biology of the target species and generate data on temporal and spatial changes. Specifically, in flying insects, wing or body shape and size can be used as a population marker to detect differences between non-resistant and resistant variants as shown for the western corn rootworm [21,23]. In CM, the effect of resistance on wing shape and size changes has not been examined. However, there have been several authors who have investigated how CM wing size, shape, and wing geometry are influenced by geography, wing speed, and elevation (e.g., CM from Iran [33,34]) and sex, location, and host plant (e.g., CM from Chile [35]). Studies on CM and insecticide resistance have shown that larger females are more resistant than smaller males [19,36,37]. However, there is a need to further investigate CM wing (shape and size) performance in relation to key environmental dispersal constraints (i.e., wind) to determine how well different wing morphotypes perform. Through the use of the finite element method (FEM), it is possible to investigate flight performance by modelling various wing attributes (shape, size, and membrane and vein thickness) in relation to environmental variables such as wind that either hinder or help CM in their short- and long-distance dispersal. During the past two decades, FEM has been used to investigate flight performance of insect species, for example, modelling flexural stiffness (wing deformation) in hawkmoth [38,39] and wing-bending asymmetry in Japanese rhinoceros beetle [40]. FEM is a powerful tool to model some aspects of insect flight behavior without the need for complicated laboratory equipment (i.e., flight mill and tethering experiments [41,42]). For another significant pest insect, western corn rootworm, Mikac et al. [23] recently discussed how the combined approach of geometric morphometrics and FEM would enable the modelling of wing structure and fight efficiency of wing shape differences based on crop resistance to produce robust data on understanding how pest insects change their wing shape and size and dispersal efficiencies under a resistance scenario. Such information is crucial to developing successful integrated resistance management strategies.

Here, we use forewing size and shape and wing deformation differences to examine changes in CM as related to the development of resistance to control practices. The consequences of changes in CM wing shape and size and wing deformation (i.e., modelling wing deformation against differing wind conditions using FEM) are discussed in relation to their long-distance flight and dispersal capabilities (i.e., possible spread of resistant alleles). Also discussed is how forewing morphology can be used as an inexpensive and accessible population biomarker for CM resistance detection globally.

## 2. Materials and Methods

### 2.1. Collection Sites and Sampling

Adult male CM were collected in 2017 in the apple orchards in the continental part of Croatia (Figure 1) by using delta sticky pheromone traps (Csalomon^®^: Plant Protection Institute, Centre for Agricultural Research Hungarian Academy of Sciences, Budapest, Hungary). This area of Croatia is characterized by a continental-humid climate of warm, rainy summers and cold winters [43]. In total, ten field populations (six from integrated and four from ecological apple production) and one laboratory-reared population (which was never exposed to insecticides), were investigated. The selected orchards were 15–20 years old, and represent typical apple management within Croatia. The distance between integrated pest management (IPM) and ecological orchards was approximately 30 km. Control activities in IPM orchards included pest and disease monitoring systems and threshold-based applications in accordance with a standard EU Directive [44]. The IPM orchard was regularly treated with a variety of pesticides, 10–15 times during the growing season. The pesticides used in the IPM orchard were applied by spraying and included the following: organophosphate insecticides (chlorpyrifos-ethil); pyrethroids (alpha-cypermethrin, deltamethrin, etc.); insect growth regulators (lufenuron, methoxyfenzoide); neonicotinoids (thiacloprid, acetamiprid); naturalites (emamectin-benzoate); and diamids (chlorantraniliprole). Reyes et al. [7,8] confirmed resistance of European CM populations to all of these insecticides in orchards in commercial apple production systems. Ecological orchards were not treated with chemical agents, and pests were controlled mainly through the maintenance of high levels of functional biodiversity (i.e., beneficial insect assemblages). CM from ecological orchards were assumed to be susceptible populations.

From each location, 20–30 individuals were collected and a total of 294 moths were analyzed. Forewings were removed from each individual. The moth’s wing scales were removed by immersion in fluorine. Cleared forewings were slide mounted using the fixing agent Euparal (Carl Roth GmbH + Co. KG, Karlsruhe, Germany) based on standard methods [45]. Slide-mounted wings were photographed using a Leica DFC295 digital camera (3 megapixel) on a trinocular mount of a Leica MZ16a stereo-microscope and saved in JPEG format using the Leica Application Suite v. 3.8.0 (Leica Microsystems Ltd.: Leica Microsystems GmbH, Wetzlar, Germany). Eighteen type 1 landmarks defined by vein junctions or vein terminations [46] were used in morphometric shape analyses (Figure 2). Morphological type 1 landmarks are points that can be located precisely on each specimen under study and are clearly discernable from specimen to specimen.

### 2.2. Geometric Morphometric Analysis

The 18 type 1 landmarks chosen were digitized using TPSDIG v. 2.17 (NY State University at Stony Brook, New York, NY, USA) [47]. The landmark coordinates data were aligned by Procrustes superimposition [48,49] using the R statistical environment v. 3.6.0 (R Foundation Statistical Computing, Vienna, Austria) [50] and the package geomorph v. 3.1.0 [51].

To test for and thus avoid measurement errors that may occur via inaccurate digitizing, a Procrustes ANOVA of 30 left or right moth forewings was undertaken. Forewings were digitized twice and values of the mean squares (MS) from the ANOVA between the error component of variation and individuals were compared using MorphoJ v. 1.06d [52,53]. In order to avoid an allometry effect on the data, a multivariate regression of shape (Procrustes coordinate) on size (centroid size) was taken into account [54]. Principal component analysis (PCA) was used to visualize forewing shape variation related to the apple production system [55]. The PCA was based on the covariance matrix of individual forewing shape. To visualize the average shape changes of the laboratory, integrated, and ecological populations, a covariance matrix of the average data was created, and the average wireframe exported. A Procrustes ANOVA with permutation procedures was performed to assess differences among the three groups describing patterns of shape variation using the Procrustes shape variables. Finally, a canonical variate analysis (CVA) with cross-validations was run using the ‘CVA’ function within the package Morpho v. 2.3.0 (R Foundation Statistical Computing: Vienna, Austria) [56] to explore the morphological shape variables that were maximized between resistant and non-resistant group variance relative to within-group variance. Within the CVA morphospace, 95% confidence intervals (CI) were generated around each of the groups.

### 2.3. Finite Element Method (FEM)

To examine deformation of forewings based on a variety of wind speeds, CM from the three treatment populations described above were investigated using FEM. Three simplified finite element models of the CM wings where tested using ANSYS Workbench v. 19 (ANSYS^®^ Academic Research Mechanical, Release 19.0, ANSYS Inc., Canonsburg, PA, USA). The CM wing model was represented as a tridimensional geometrical model of a round solid bar as a simulation of the real model based on the wireframe of the average shape produced as detailed above. For all vein elements, the material properties were characterized as being isotropic linear elastic materials with a density of 1200 kg m^−3^, thickness of 45 µm, Poisson’s ratio of 0.3, and Young’s modulus of 150 Mpa [38]. The mesh analysis of the vein structure included 3161 quadratic elements that proved to be enough to ensure asymptotic performance of the model. Several ranges of incremental loads were applied along the length of the wireframes from the dorsal side (*z* axis) to represent the wind force applied to the CM wings under field conditions. Such incremental loads were from 2.8 m/s^−1^ up to 27.8 m/s^−1^ as a representation of the lift force for a CM wing under steady flight. In order to represent the connection of the wireframe to the body of the CM, a fixed support was applied to the position of landmark number 18 of the wireframe in this model (Figure 1).

## 3. Results

### 3.1. Geometric Morphometrics

The MS of Procrustes ANOVA for individual variation (MS = 0.000114) exceeded the error (MS = 0.00000523). The multivariate regression showed considerably smaller values of size influence on the data, therefore, a correction was not necessary for the data (prediction of allometric influence: 4.92%). A PCA of the covariance matrix of the individual shape showed that the first three dimensions of the shape space accounted for 43% of the total variance (PC1 = 17.1%; PC2 = 13.4%; PC3 = 12.5%). Wireframes of the average shape among groups were exported and superimposed showing a clear pattern of variation among landmarks 7, 8, and 9 from the tip of the basal veins and landmarks 1 and 12 that resulted in elongated and widened wings (Figure 3). Significant differences in shape variation were clearly evident among treatment groups after running 10,000 permutations from the Procrustes ANOVA (Table 1).

CVA (Figure 4) showed significantly different groupings within the morphospace (*p* < 0.0001), on the extreme negative of canonical function 1 (CV1, red dots). This region of morphospace was characterized by moths with the most elongated wing shape of all three production groups, resulting from a larger expansion of landmarks 8 and 9 (Figure 4B). The remaining groups, however, occupied opposite extremes of CV2 (CV2, green and black dots) and a similar wing shape variation was found. However, these individuals exhibited more variation related to the structural portion of the wing for landmarks 16 resulting in an elongated wing by the movements of the landmarks 7 and 8 (Figure 4A,C).

### 3.2. Finite Element Method

For the three CM wireframes (ecological, integrated, and laboratory-reared) FEM models indicated that deformation of the veins was directly related to the geometry and material properties of the wing. As wind speed (m/s^−1^) increased, so too did wing deformation (mm) for CM from each of the three treatments modelled (Figure 5). CM from the ecological orchards displayed the least deformation of 32% at a maximum wind speed of 4.2 m/s^−1^, followed by integrated CM with 32.3%, and the laboratory-reared with 35.7% wing deformation.

These results illustrate that laboratory-reared CM were more sensitive to larger deformations than the ecological and the integrated CM wings (Figure 5). This larger deformation affects CM wing structure to produce a buckling effect on the wing. It was found that wing structure could not withstand loads with its original shape, so that it changed shape to find a new equilibrium configuration. This is an undesirable process and occurred for a well-defined value of the load. The consequences of buckling were geometric and potentially affected CM flight (Figure 6, Figure 7 and Figure 8).

## 4. Discussion

The present study is the first to focus on investigating wing morphology of CM in relation to different control practices (integrated and ecological orchard management).

For several decades, insecticide applications have been the dominant tools used for CM control [57], and in Europe where resistance in known, only a few registered insecticides are still effective [7]. Resistance mechanisms in CM are numerous [7,58] and bioassays for resistance detection in field populations is time consuming and expensive. Nevertheless, because of adverse effects of insecticide use to human health and the greater environment [57], an area wide-IPM (AW-IPM) approach was adopted for CM control worldwide. AW-IPM is mainly based on the use of pheromones (e.g., mating disruption: [57,59]) but where conditions for its effective application are suboptimal, additional treatments for pest control need to be applied [60], representing a potential threat to the re-emergence of resistance. Therefore, the regular monitoring for insecticide resistance is essential in order to proactively prevent insecticide resistance from compromising control.

In this study, we showed a reliable pattern of differences in forewing shape related to orchard control practice type. Findings support previous work where phenotypic differences in wing shape and size were shown to be reliable population biomarkers [21,22,23,27,28,29,61,62]. From the data gathered in this study, it is clear that morphological differences are present in the forewing shape of CM based on control practices (ecological vs. integrated). Differences in wing shape between laboratory and field populations were investigated to determine if rearing conditions were related to the developmental stability of the organism (as suggested by Gerard et al. [63]). The differences found between ecological and integrated moth populations suggest that rearing conditions were not the main reason for observed differences.

In order to prevent the spread of resistant CM populations in the field, it is crucial to detect specific biotypes in a timely manner to apply effective control measures. If the frequency of resistance alleles builds up unchecked, resistance may eventually become ‘fixed’ in populations. Once resistance reaches very high levels, strategies to restore susceptibility are unlikely to be effective. Many authors agree that only by monitoring, characterizing, and predicting the appearance and spread of resistance can we hope to use existing chemical tools in a sustainable manner [64,65,66]. Therefore, there is a real need for alternative tools such as geometric morphometrics (GM) that can be used to more easily detect resistance in field-collected populations and from which a decision about IPM can be made.

By applying GM analyses, detection of different biotypes of some other insects such as spotted wing drosophila [62], western corn rootworm [21,23], ground beetle [61], and beet root weevil [29] was confirmed as a diagnostic tool and consequently introduced into IPM practice. Here, field populations were investigated in detail by orchard control type, and differences in shape variation were found for CM. The forewings of moths from ecological orchards were significantly more elongated in shape and narrower in width compared with moths from IPM orchards. Identical results were found for corn rootworm [23] where the same venation patterns were observed in comparison to resistant and non-resistant beetles. There is now more evidence from multiple insect taxa that resistance can influence wing shape in insects; with this influence being easier to detect on a phenotypic than genetic level [26]. Mikac et al. [23] suggested that such phenotypic differences have an impact on dispersal and long-distance movement of resistant and non-resistant insects because wing morphology is a critical element of an insect’s dispersal capacity (also see [67]). Finding the wing morphotype of superior fliers and dispersers, as well as morphotypes with specific resistance patterns, is crucial to the success of resistant pest management strategies. Although Mikac et al. [21,22,23] investigated and demonstrated that wing shape differences exist in resistant and non-resistant corn rootworm beetles, this is the first study to demonstrate evidence of distinct lepidopteran wing shape differences related to resistance.

Further investigation of CM wing changes related to control practices using FEM indicated that the deformation of CM vein structure was directly related to the geometry and properties of the materials of the wings themselves. FEM of CM from the ecological population showed that those individuals had the least wing deformation at the highest wind speeds modelled. It is likely that this wing/vein pattern was stronger in its structural integrity and therefore rendered CM from ecological populations better fliers and, by extension, dispersers. The biological importance of this is that CM from ecological populations and potentially those susceptible to resistance were most likely to spread their alleles through the landscape (cf. corn rootworm [23]). Of particular interest were CM from integrated populations that were suspected to be resistant to various chemical treatments, which through FEM were shown to have similar wing deformations to those of ecological (or susceptible) populations when modelled at the highest wing speeds. This result seems to indicate that from IPM orchards, CM are being exposed to selection pressures that result in a phenotype better able to withstand higher wing speeds than laboratory-bred populations. There is a possibility that both control measures (ecological and IPM) are having no net difference on CM’s capacity to fly (as evidenced: Figure 5), despite the difference in overall wing shape and size shown using GM. Clearly there is a need for a more thorough understanding of whether control practices are effective at curtailing CM’s ability to disperse; for CM dispersal under maximum wind conditions, this could be up to 11 km/day [41,68]. Further FEM modelling is required to examine in more detail how CM wing structural integrity changes according to differing control practices and how this might impact upon their ability to spread resistant alleles through a pome fruit agricultural landscape. This will enable a deeper understanding of flight and dispersal capabilities of CM, which is fundamental to informing IPM strategies for this pest. A limitation of this study was not knowing to which compounds CM were resistant to, and also the intensity of resistance. Therefore, future work will focus on testing for specific resistant strains of CM and investigating GM and FEM differences for each of these.

## 5. Conclusions

Overall implications from this work suggest that GM techniques can be used to detect population changes related to different types of apple production and could serve as alternative biomarkers to more expensive and specialized-use genetic markers when investigating field resistance. Coupled with the use of FEM to model how wings respond to various wind conditions, GM will enable a deeper understanding of how resistant alleles are spread through the landscape. The detection and monitoring of resistant variants is the first step towards the implementation of anti-resistant strategies and the sustainable use of pesticides in apple production worldwide when controlling CM.

## Figures and Tables

**Figure 1 insects-10-00310-f001:**
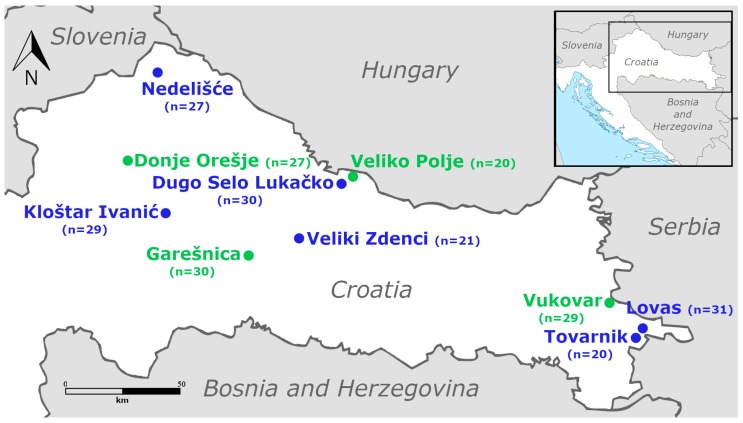
Sampling sites of codling moth in Croatian orchards (blue: integrated pest management (IPM) orchard, green: ecological orchard; n = number of individuals per location).

**Figure 2 insects-10-00310-f002:**
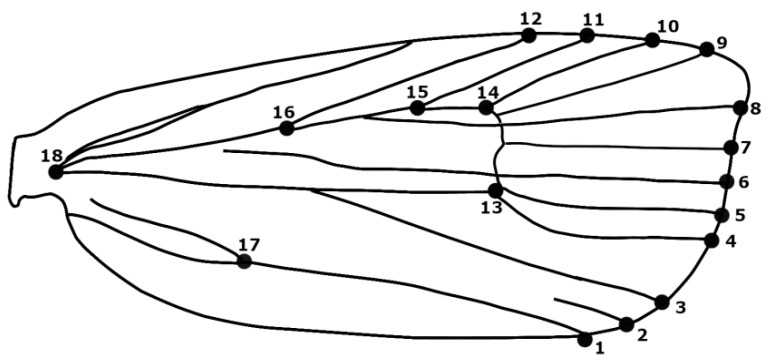
Position of 18 type 1 landmarks on a codling moth forewing.

**Figure 3 insects-10-00310-f003:**
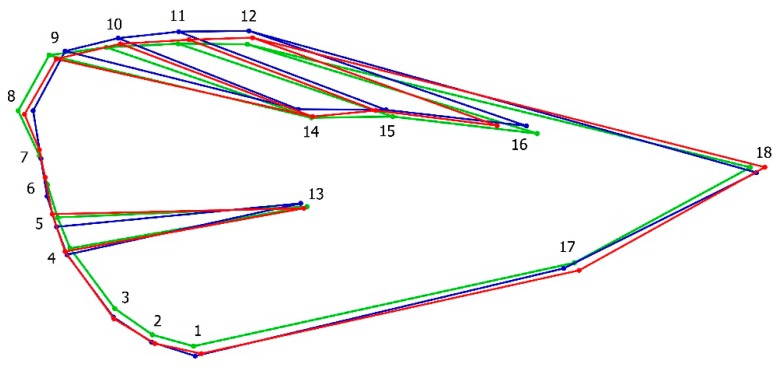
Wireframe visualization of the average shape for different management types of codling moth: ecological (green); laboratory-reared (red); integrated (blue) populations.

**Figure 4 insects-10-00310-f004:**
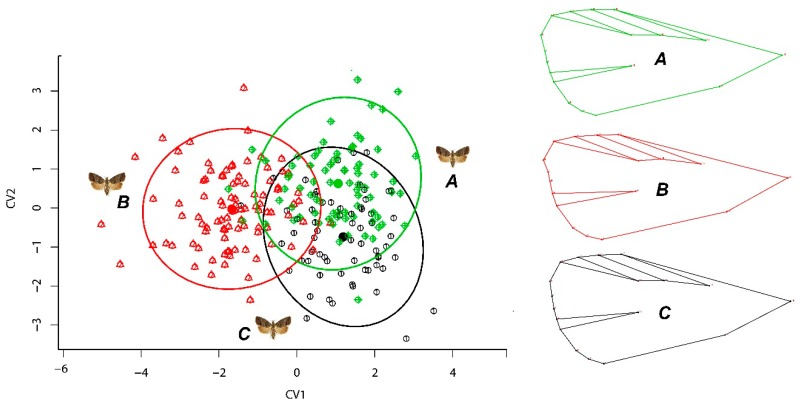
Canonical variate analysis (CVA) visualization of forewing shape differences for codling moth: ecological (green, **A**); laboratory-reared (red, **B**); integrated (black, **C**).

**Figure 5 insects-10-00310-f005:**
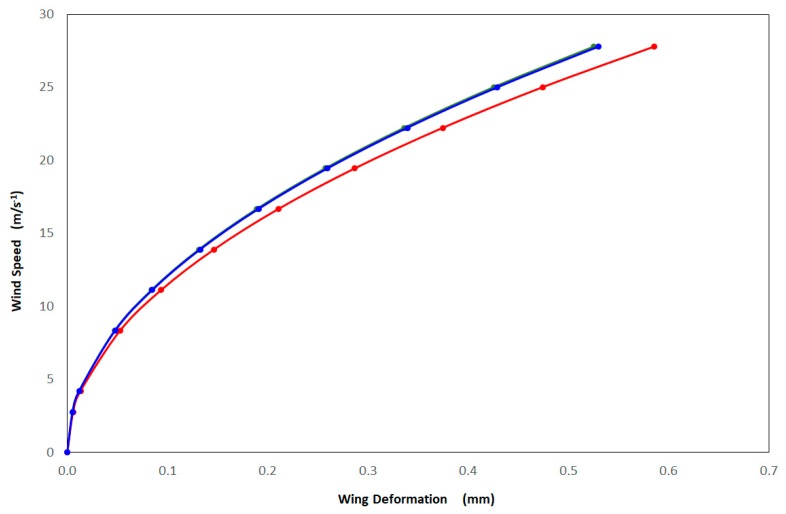
Finite element method-modelled maximum wing deformations (mm) versus wind speed (m/s^−1^) of the three codling moth treatment populations: ecological (green); integrated (blue); and laboratory-reared (red).

**Figure 6 insects-10-00310-f006:**
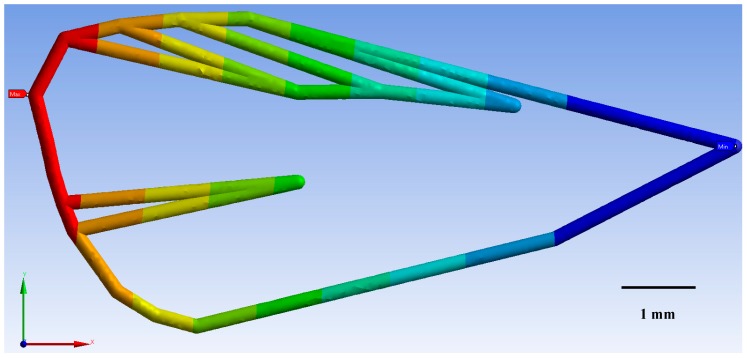
Codling moth from the ecological populations exhibited a maximum deformation (red area) of 0.0118 mm around landmark 8 (Figure 1) for a maximum wind speed of 4.2 m/s^−1^. The minimal deformation of 0.0013 mm was concentrated around landmark 17 (light blue area on figure).

**Figure 7 insects-10-00310-f007:**
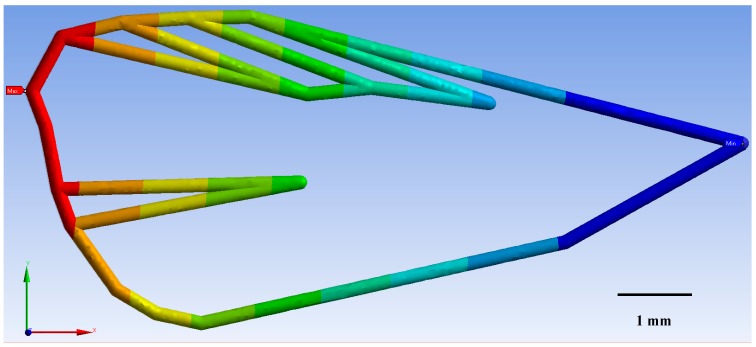
Codling moth from the laboratory-reared population exhibited a maximum deformation (red area) of 0.013 mm around landmark 8 (Figure 1), for a maximum wind speed of 4.2 m/s^−1^. The minimal deformation of 0.0014 mm was concentrated around landmark 17 (light blue area on figure).

**Figure 8 insects-10-00310-f008:**
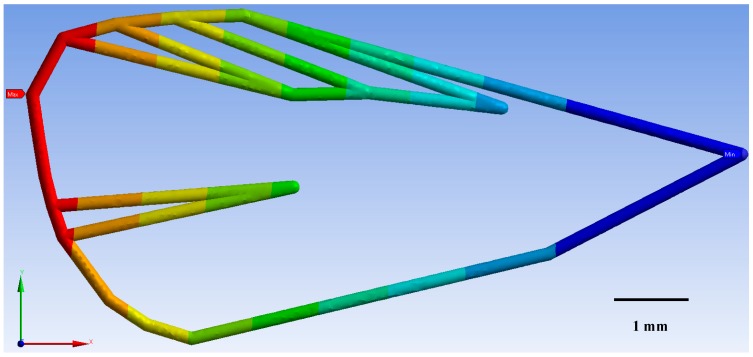
Codling moth from the integrated population exhibited a maximum deformation (red area) of 0.0112 mm around landmark 8 (Figure 1), for a maximum wind speed of 4.2 m/s^−1^. The minimal deformation of 0.0013 mm was concentrated around landmark 17 (light blue area on figure).

**Table 1 insects-10-00310-t001:** Procrustes ANOVA for both centroid size and shape of codling moth populations.

	df	SS	MS	Rsq	F	Z	Pr (>F)*
Resistant Type	2	0.01	0.007	0.04	55.74	56.63	0.001 **
Sex	1	0.007	0.007	0.02	54.18	43.41	0.001 **
Residuals	235	0.31	0.001	0.92			
Total	238	0.33					

*df = degrees of freedom; SS = sum of squares; MS = mean squares; Rsq = R squared; F = F ratio; Z = Z score; Pr(>F) = *p*-value associated with the F statistic; ** statistically highly significant

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
