# Peer review of "Codling Moth Wing Morphology Changes Due to Insecticide Resistance"

_insects, 2019, doi:10.3390/insects10100310_

Round 1

Reviewer 1 Report

In their study on the morphometrics of codling moth (Cydia pomonella) wings in response to different management types Živković et al. present an interesting case study that demonstrates the utility of morphometrics for use as a bio-marker for distinguishing potentially resistant vs. potentially susceptible strains of an important insect pest.  I thoroughly enjoyed reading the manuscript, and look forward to sharing their approach with colleagues after its publication.  In general I found the manuscript well written, the methods and results clear and easy to follow, and the discussion did a good job of presenting the findings in relation to other studies.  I have only minor comments in regards to clarifications in the text.

1) could "ecological management" be defined upon its first mention?

2) similar, could "integrated" be followed by "IPM" after its first mention?

3) Could the authors explain why most, but not all, of the vein junctions and terminations were chosen as landmarks?  As is, being that the vast majority were chosen, it's raises concerns that the others were not?  A sentence could most likely explain how junctions/terminations were selected.

4) The main result of ecological (and thus likely susceptible) strains having wings more resistant to deformation than the wings of IPM (and thus likely resistant) strains needs a bit more clarification.  As is, it seems like the IPM methods are actually helpful in that they are negatively-deforming the wings.  Is that true?

5) building on that, it might be helpful to add a sentence that clearly states that these are not susceptible and resistant strains, per se, but rather from orchards where the development of resistance is or is not likely (or something to that effect).

Author Response

13th September 2019

Dear Dr Belegisanin,

Please find below our detailed response to the reviewers’ comments.

We thank you, the subject editor and the reviewers for their considered and helpful comments and look forward to working further with you to have our manuscript published in Insects.

Sincerely,

Dr Darija Lemic

Assistant Professsor

University of Zagreb, Faculty of Agriculture

Department for Agricultural Zoology

25 Svetosimunska Street

Zagreb 10 000

Croatia

Reviewer #1:

In their study on the morphometrics of codling moth (Cydia pomonella) wings in response to different management types Živković et al. present an interesting case study that demonstrates the utility of morphometrics for use as a bio-marker for distinguishing potentially resistant vs. potentially susceptible strains of an important insect pest.  I thoroughly enjoyed reading the manuscript, and look forward to sharing their approach with colleagues after its publication.  In general I found the manuscript well written, the methods and results clear and easy to follow, and the discussion did a good job of presenting the findings in relation to other studies.  I have only minor comments in regards to clarifications in the text.

Q1. 1) could "ecological management" be defined upon its first mention?

Response: Done.

Q2. 2) similar, could "integrated" be followed by "IPM" after its first mention?

Response: Done.

Q3. 3) Could the authors explain why most, but not all, of the vein junctions and terminations were chosen as landmarks?  As is, being that the vast majority were chosen, it's raises concerns that the others were not?  A sentence could most likely explain how junctions/terminations were selected.

Response: Normally in moths the selection of the vein intersections occurs because it most closely follows the definition of a type 1 landmark that can be reliably and repeatedly located among individuals (ie. anatomical homology). However, a factor that always influences the final decision as to keep or omit particular landmarks is based on the quality of the sample and ability to successful slide mount specimens. This is not always an easy feat and certainly at times will dictate which landmarks are used in final analysis.  Here we have done our best to still retain enough landmarks so as to not compromise the statistical power of analyses.

Q4. 4) The main result of ecological (and thus likely susceptible) strains having wings more resistant to deformation than the wings of IPM (and thus likely resistant) strains needs a bit more clarification.  As is, it seems like the IPM methods are actually helpful in that they are negatively-deforming the wings.  Is that true?

Response: Additional clarification is now provided in the discussion and a change in the scale of figure 5 (to better explore how wind speed impacts wing deformation) has shown that IPM measures are similar to CM from ecological orchards. This result seems to indicate that from IPM orchards CM are being exposed to selection pressures that result in a phenotype better able to withstand higher wing speeds than laboratory bred populations. There is a possibility that both control measures (ecological and IPM) are having no net difference on their capacity to disperse (as evidenced by Fig 5), despite the difference in overall wing shape and size shown using GM. This sentence and clarification now appears in the discussion.

Q5. 5) building on that, it might be helpful to add a sentence that clearly states that these are not susceptible and resistant strains, per se, but rather from orchards where the development of resistance is or is not likely (or something to that effect).

Response: Lines 136-144 clearly state that CM from IPM orchards are likely to have resistance to the chemicals used to control them. Line 45 now addressed that ecological populations in this study are susceptible. 

Reviewer 2 Report

Comments on the manuscript: Codling moth wing morphology changes due to 2 insecticide resistance by Pajač Živković1 et al.

 Codling moth is one of the main pests of apples in fruit production areas in Europe and many areas of the world. A strict cosmetic requirement causes a high use of insecticides, and other IPM tools including mating disruption to manage codling moth. A tool to discriminate resistance populations are very important for an IRM. This manuscript provides very good information about the wing morphology and functionality in the sampled populations. The methods and analysis to separate the morphology in different strains seemed to be adequate. However, and additional analysis to compare between individuals of the same orchard and between orchards are critical. I believe this information is important to publish. However, there are some limitations on the conclusions of this research, and they should be addressed. Please see my comments below. 

L100-104 Rephrase. If smaller individuals are susceptible, they are not carrying resistant genes.

L125 Provided the number of individuals analyzed from each population and each orchard.  

L41-41 Reyes found resistance of CM in European populations in commercial orchards. Please provide specific data of resistance on those IPM orchards where moths were collected for the morphology study. This is very important for the discussion.

If there are morphology differences between the IPM versus ecologically managed orchards this method only discriminates between the type of management, but not to which compounds codling moth is resistant, and the intensity of resistance. In addition, other forces might be responsible for the differences in wing morphology. I would expect a high difference in fruit pest infestation in IPM versus ecologically manage orchard and then more selection of individuals more adapted to disperse to less compete niches. Are the authors are considering using laboratory selected population to see if insecticide selection is responsible for wing morphology? 

L139 The correct spelling is methoxyfenozide

Figure 4. This figure showed the data points of the sampled moth. However, there are some overlapping of the three strains of the forewing shape.  I wonder if there are differences between individuals in the same orchard. Therefore, comparison in individuals from the same orchard is very important.  In addition, since there are data from all individuals in all the orchards would be critical to make the separation by orchard regardless of the type of collection (IMP, ecologically managed, or lab). It seems to be very low the number of collected and analyzed moths per orchard.

Is it possible to add a standard error in Figure 5, or intervals (upper and lower limit)? Separate the morphs (suspected resistant-IPM or susceptible) base on intervals.

Finally, this study didn’t show a direct cause and effect-if the wing changes is directly related to use of pesticides. Other factor might intervene in this change, or perhaps just one insecticide was responsible for the changes. Please address the limitations of this study.

Author Response

13th September 2019

Dear Dr Belegisanin,

Please find below our detailed response to the reviewers’ comments.

We thank you, the subject editor and the reviewers for their considered and helpful comments and look forward to working further with you to have our manuscript published in Insects.

Sincerely,

Dr Darija Lemic

Assistant Professsor

University of Zagreb, Faculty of Agriculture

Department for Agricultural Zoology

25 Svetosimunska Street

Zagreb 10 000

Croatia

Reviewer #2:

Comments on the manuscript: Codling moth wing morphology changes due to insecticide resistance by Pajač Živković et al.

Codling moth is one of the main pests of apples in fruit production areas in Europe and many areas of the world. A strict cosmetic requirement causes a high use of insecticides, and other IPM tools including mating disruption to manage codling moth. A tool to discriminate resistance populations are very important for an IRM. This manuscript provides very good information about the wing morphology and functionality in the sampled populations. The methods and analysis to separate the morphology in different strains seemed to be adequate. However, and additional analysis to compare between individuals of the same orchard and between orchards are critical. I believe this information is important to publish. However, there are some limitations on the conclusions of this research, and they should be addressed. Please see my comments below. 

Q1. L100-104 Rephrase. If smaller individuals are susceptible, they are not carrying resistant genes.

Response: This part has been rewritten and clarified.

Q2. L125 Provided the number of individuals analyzed from each population and each orchard.  

Response: The number of individuals analyzed from each location has been added into the Material and Methods sections, and n been added into Figure 1.

Q3. L41-41 Reyes found resistance of CM in European populations in commercial orchards. Please provide specific data of resistance on those IPM orchards where moths were collected for the morphology study. This is very important for the discussion.

Response: please see response directly next as it relates to the response for this comment too.

Q4. If there are morphology differences between the IPM versus ecologically managed orchards this method only discriminates between the type of management, but not to which compounds codling moth is resistant, and the intensity of resistance. In addition, other forces might be responsible for the differences in wing morphology. I would expect a high difference in fruit pest infestation in IPM versus ecologically manage orchard and then more selection of individuals more adapted to disperse to less compete niches. Are the authors are considering using laboratory selected population to see if insecticide selection is responsible for wing morphology? 

Response: According to IRAC for codling moth control in most countries there are 11 modes of action available to the market depending on the country. For CM there are insecticides that affect their nervous system and pest growth and development. Acetylcholinesterase inhibitors (carbamates and organophosphates), sodium channel modulators (pyrethroids), nicotinic acetylcholine receptor agonists (neonicotinoides), nicotinic acetylcholine receptor agonists allosteric modulators (spinosyns), chloride channel activators (avermectins), voltage dependent sodium channel blockers (oxadiazines) and ryanodine receptor modulators (diamides) affect pest nervous systems. They act on nerve and muscle targets, generally these insecticides are fast acting. In our IPM orchards all insecticide treatments targeted the nervous system (OP, pyrethroids and neonicotinoids….) and we assume that CM developed resistance to that mode of action. We are aware that we do not tested specific CM resistant type, and these limitations have been discussed in our manuscript.

Q5. L139 the correct spelling is methoxyfenozide.

Response: Change in spelling made.

Q6. Figure 4. This figure showed the data points of the sampled moth. However, there are some overlapping of the three strains of the forewing shape.  I wonder if there are differences between individuals in the same orchard. Therefore, comparison in individuals from the same orchard is very important.  In addition, since there are data from all individuals in all the orchards would be critical to make the separation by orchard regardless of the type of collection (IMP, ecologically managed, or lab). It seems to be very low the number of collected and analyzed moths per orchard.

Response: The CVA figure shows overlapping points and this means that the shape variation of some month wings have similarities within the population investigated. It is well within expected results when spatially proximate populations are investigated that that appear closer in the CVA output.  Although, intrapopulation differentiation can be detected it was not part of the aims of this research. Intra-morphological differences are a genetic aspect of population variation and beyond the scope of the research we conducted in this paper.

Q7. Is it possible to add a standard error in Figure 5, or intervals (upper and lower limit)? Separate the morphs (suspected resistant-IPM or susceptible) base on intervals.

Response: Each curve is the result of the modelled wing deformation according to wing speed. Figure 5 and the analysis it represents is a simulation (finite element method). Using FEM, displacement is measured by modelling the deformation (in mm) of a structure (here, the wing) versus an applied force (here, wind speed). It is not a regression analysis, therefore standard error is not appropriate.

Q8. Finally, this study didn’t show a direct cause and effect-if the wing changes is directly related to use of pesticides. Other factor might intervene in this change, or perhaps just one insecticide was responsible for the changes. Please address the limitations of this study.

Response: The limitations of this study and plans for future investigation have been included in the manuscript at the end of the discussion.

Round 2

Reviewer 2 Report

Authors had addressed the comments and included the possible limitations of the research. This paper should be published. The researchers did an excellent job in the metrics and analysis of wing morphology.